# Local Differential Privacy for Mixtures of Experts

## Abstract

We introduce a new approach to the mixture of experts model that consists in imposing local differential privacy on the gating mechanism. This is theoretically justified by statistical learning theory. Notably, we provide generalization bounds specifically tailored for mixtures of experts, leveraging the one-out-of-$n$ gating mechanism rather than the more common $n$-out-of-$n$ mechanism. Moreover, through experiments, we show that our approach improves the generalization ability of mixtures of experts.

## 1 Introduction

Mixtures of experts, initially introduced by Jacobs et al. [1991], have found widespread use in modeling sequential data, including applications in classification, regression, pattern recognition and feature selection tasks (Städler et al. [2010] and Khalili and Lin [2013]). One of the fundamental motivations behind mixtures of experts is their ability to break down complex problems into more manageable sub-problems, potentially simplifying the overall task. The structure of these models is well suited to capturing unobservable heterogeneity in the data generation process, dealing with this problem by splitting the data into homogeneous subsets (with the gating network) and associating each subset with an expert. This intuitive architecture has led to significant interest in mixture of experts models, resulting in a wealth of research (Yuksel et al. [2012]), ranging from simple mixtures of experts ( Jacobs et al. [1991], Jordan and Jacobs [1993]) to sparsely gated models (Shazeer et al. [2017]). Moreover, this architecture has inspired the development of various other models, such as switch transformers (Fedus et al. [2022]). However, despite the considerable attention mixtures of experts have received, advancements in their theoretical analysis have been relatively limited. Azran and Meir [2004] proved data-dependent risk bounds for mixtures of experts (with the $n$-out-of-$n$ gating mechanism) using Rademacher complexity, but they exhibit a dependence on the complexity of the class of gating networks and the sum of the complexities of the expert classes, which reflects the complex structure of mixtures of experts but unfortunately leads to potentially large bounds. We are not aware of other work proving generalization bounds specifically tailored to mixtures of experts.

To make theoretical progress, we utilize a well-known privacy-preserving technique called Local Differential Privacy (LDP). It was initially introduced by Dwork [2006] and has since been widely used to preserve privacy for individual data points as in Kasiviswanathan et al. [2010]. This is achieved by introducing stochasticity in algorithm outputs to control their dependence on specific inputs. This stochasticity is generally quantified by a positive real number $\epsilon$. In this case, we write $\epsilon$-LDP instead of just LDP. The parameter $\epsilon$ quantifies the level of privacy protection in the local differential privacy mechanism. A smaller value indicates stronger privacy protection, which requires the addition of more noise.

In this work, we exploit this noise for regularization in our models by imposing the $\epsilon$-LDP condition on their gating networks. This method allows us to leverage the numerous benefits of the most complex architectures, such as neural networks, without compromising theoretical guarantees on risk. By relying on LDP, we offer tight theoretical guarantees on the risk of mixtures of experts models,

39  provided with the one-out-of-$n$ gating mechanism. Unlike the very few existing guarantees, these
40  bounds depend only logarithmically on the number of experts we have, and the complexity of the
41  gating network only appears in our bounds through the parameter $\epsilon$ of the LDP condition.

## 2  Preliminaries

43  Let $\mathcal{X}$ be the instance space, $\mathcal{Y}$ the label space, and $\mathcal{Y}'$ the output space (which can be different
44  from $\mathcal{Y}$). As is usual in supervised learning, we assume that data $(x, y) \in \mathcal{X} \times \mathcal{Y}$ are generated
45  independently from an unknown probability distribution $\mathcal{D}$. We consider a training set of $m$ examples
46  $S = ((x_1, y_1), \dots, (x_m, y_m)) \sim \mathcal{D}^m$ and a bounded loss function $\ell \colon \mathcal{Y}' \times \mathcal{Y} \to [0, 1]$.

### 2.1  Mixtures of experts

48  We consider classes $\mathcal{H}_i$ of experts $h_i \colon \mathcal{X} \to \mathcal{Y}'$ for $i = 1, \dots, n$. Let $\mathcal{G}$ be a set of gating functions
49  $\mathbf{g} \colon \mathcal{X} \to [0, 1]^n$ such that, given any $x \in \mathcal{X}$, we have that $\sum_{i=1}^n g_i(x) = 1$, where $g_i(x)$ is the
50  $i$-th component of $\mathbf{g}(x)$. This means that each gating function defines a probability distribution on
51  $[n] = \{1, \dots, n\}$ for each $x \in \mathcal{X}$, where $g_i(x)$ is the probability of $i$.

52  In this work, a mixture of experts consists of $n$ experts, $\mathbf{h} = (h_1, \dots, h_n) \in \mathcal{H}_1 \times \cdots \times \mathcal{H}_n$, a
53  gating function $\mathbf{g} \in \mathcal{G}$ and a gating mechanism that combines the outputs of the experts and the
54  output of the gating function to produce the final output. Our models use the stochastic one-out-of-$n$
55  gating mechanism, as described in Jacobs et al. [1991]. It is defined as follows: to make a prediction
56  with $(\mathbf{g}, \mathbf{h}) \in \mathcal{G} \times \prod_{i=1}^n \mathcal{H}_i$ given an instance $x$, draw $i \sim \mathbf{g}(x)$ and output $h_i(x)$. This stochastic
57  predictor has risk and empirical risk defined by, respectively,

$$R(\mathbf{g}, \mathbf{h}) = \mathop{\mathbb{E}}_{(x,y) \sim \mathcal{D}} \mathop{\mathbb{E}}_{i \sim \mathbf{g}(x)} \ell(h_i(x), y), \quad \text{and} \quad R_S(\mathbf{g}, \mathbf{h}) = \frac{1}{m} \sum_{j=1}^m \mathop{\mathbb{E}}_{i \sim \mathbf{g}(x_j)} \ell(h_i(x_j), y_j).$$

58  The preference for the one-out-of-$n$ gating mechanism over the $n$-out-of-$n$ mechanism in mixtures
59  of experts is justified by its ability to induce sparsity and noise, enhancing computational efficiency
60  and robustness to overfitting. This sparsity also offers scalability benefits, particularly in large-scale
61  applications, where activating all experts for each input can lead to increased computational and
62  memory requirements as explained in Shazeer et al. [2017] and Jacobs et al. [1991]. Moreover, the
63  one-out-of-$n$ mechanism is more amenable to certain kinds of theoretical analysis, including ours.

### 2.2  Local Differential Privacy

65  **Definition 2.1.** Let $\mathcal{I}$ be a finite set, consider a mechanism that produces an output $i \in \mathcal{I}$, given an
66  input $x \in \mathcal{X}$, with probability $\mathbb{P}(i \,|\, x)$, and let $\epsilon$ be a nonnegative real number. Then, the mechanism
67  satisfies the $\epsilon$-Local Differential Privacy ($\epsilon$-LDP) property if and only if

$$\mathbb{P}(i \,|\, x) \le e^{\epsilon} \, \mathbb{P}(i \,|\, x') \quad \text{for all } x, x' \in \mathcal{X} \text{ and all } i \in \mathcal{I}.$$

Unless stated otherwise, we assume that each $\mathbf{g} \in \mathcal{G}$ satisfies $\epsilon$-LDP, for some fixed nonnegative real
number $\epsilon$. Since we can interpret $\mathbf{g}$ as a random mechanism that, given $x \in \mathcal{X}$, selects $i \in [n]$ with
probability $g_i(x)$, the condition of $\epsilon$-LDP amounts to the following:

$$g_i(x) \le e^{\epsilon} g_i(x') \quad \text{for all } x, x' \in \mathcal{X} \text{ and all } i \in [n].$$

68  Since $\epsilon$-LDP is an important condition for all of our theoretical results, we provide a practical way
69  of obtaining gating functions satisfying $\epsilon$-LDP from an arbitrary set $\mathcal{F}$ of bounded functions, in the
70  form of the following theorem.

71  **Theorem 2.2.** *Let $b > 0$ and $\beta \ge 0$ be real numbers, and suppose that $\mathcal{F}$ is a set of functions*
72  $\mathbf{f} \colon \mathcal{X} \to [-b, b]^n$. *Let $\mathcal{G}$ be the set of functions $\mathbf{g} \colon \mathcal{X} \to [0, 1]^n$ defined by*

$$g_i(x) = \frac{\exp(\beta f_i(x) + c_i)}{\sum_{k=1}^n \exp(\beta f_k(x) + c_k)}, \quad \text{where } \mathbf{f} = (f_1, \dots, f_n) \in \mathcal{F} \text{ and } (c_1, \dots, c_n) \in \mathbb{R}^n.$$

73  *Then, each $\mathbf{g} \in \mathcal{G}$ satisfies $4\beta b$-LDP.*

74  *Proof.* The proof is obtained by performing simple calculations, bounding the ratio $g_i(x)/g_i(x')$, for
75  all $x, x' \in \mathcal{X}$ and all $i \in [n]$. The detailed proof is given in Appendix A. $\qquad\square$

## 3 PAC-Bayesian bounds for mixtures of experts

To apply the PAC-Bayes theory, we need to add a level of stochasticity to our predictors: instead of training experts $h_i$, we train probability measures $Q_i$ on each expert set $\mathcal{H}_i$. For convenience, we write $Q = Q_1 \otimes \cdots \otimes Q_n$. Now, putting everything together, a mixture of experts $(\mathbf{g}, Q)$ makes predictions as follows: given $x \in \mathcal{X}$, draw $i \sim \mathbf{g}(x)$, then draw $h \sim Q_i$, and finally output $h(x)$. Such a predictor has risk and empirical risk defined by, respectively,

$$R(\mathbf{g}, Q) = \mathop{\mathbb{E}}_{\mathbf{h} \sim Q} R(\mathbf{g}, \mathbf{h}) \quad \text{and} \quad R_S(\mathbf{g}, Q) = \mathop{\mathbb{E}}_{\mathbf{h} \sim Q} R_S(\mathbf{g}, \mathbf{h}).$$

Notice that, though probability distributions have replaced the individual experts, there is no need to define a probability distribution on the gating functions to get a PAC-Bayesian bound. Training a single gating function will do, and, remarkably, Lemma 3.1 below shows that it can be obtained from a very complicated function, such as a neural network, provided we impose $\epsilon$-LDP (for example, with Theorem 2.2).

Finally, let us recall the notion of *Kullback-Leibler (KL) divergence*. Given probability distributions $Q_i$ and $P_i$ on $\mathcal{H}_i$, it is defined by

$$\mathrm{KL}(Q_i \,\|\, P_i) = \begin{cases} \mathop{\mathbb{E}}_{h \sim Q_i} \ln \dfrac{dQ_i}{dP_i}(h) & \text{if } Q_i \ll P_i \\ \infty & \text{otherwise,} \end{cases}$$

where $dQ_i/dP_i$ is a Radon-Nikodym derivative.

**Lemma 3.1.** *We consider mixtures of experts as defined in section 2.1 and provided with the one-out-of-$n$ routing mechanism. Let $\Delta : \mathbb{R}^2 \to \mathbb{R}$ be a convex function that is decreasing in its first argument and increasing in its second argument, and let $\epsilon$ be a nonnegative real number. Then, for any $\mathbf{g} \in \mathcal{G}$ that satisfies the $\epsilon$-LDP property, for any $Q = Q_1 \otimes \cdots \otimes Q_n$ on $\mathcal{H}_1 \times \cdots \times \mathcal{H}_n$, and for any $x' \in \mathcal{X}$:*

$$\Delta\big(e^\epsilon R_S(\mathbf{g}, Q), e^{-\epsilon} R(\mathbf{g}, Q)\big) \leq \mathop{\mathbb{E}}_{i \sim \mathbf{g}(x')} \Delta\big(R_S(Q_i), R(Q_i)\big)$$

*where $R(Q_i) = \mathbb{E}_{x \sim \mathcal{D}} \mathbb{E}_{h \sim Q_i} \ell(h(x), y)$ and $R_S(Q_i) = \frac{1}{m} \sum_{j=1}^m \mathbb{E}_{h \sim Q_i} \ell(h(x_j), y_j)$.*

*Proof.* Since the gating function satisfies $\epsilon$-LDP, we have that $e^{-\epsilon} g_i(x') \leq g_i(x) \leq e^\epsilon g_i(x')$ for all $x, x' \in \mathcal{X}$ and all $i \in [n]$. It follows that $e^\epsilon R_S(\mathbf{g}, Q) \geq \mathbb{E}_{i \sim \mathbf{g}(x')} R_S(Q_i)$ and $e^{-\epsilon} R(\mathbf{g}, Q) \leq \mathbb{E}_{i \sim \mathbf{g}(x')} R(Q_i)$. Given that $\Delta$ is decreasing in its first argument and increasing in its second argument, we find that

$$\Delta\big(e^\epsilon R_S(\mathbf{g}, Q), e^{-\epsilon} R(\mathbf{g}, Q)\big) \leq \Delta\Big(\mathop{\mathbb{E}}_{i \sim \mathbf{g}(x')} R_S(Q_i), \mathop{\mathbb{E}}_{i \sim \mathbf{g}(x')} R(Q_i)\Big)$$

Since $\Delta$ is a convex function, we can apply Jensen's inequality to the expression on the right-hand side, yielding the desired result. $\qquad\square$

Different choices of function $\Delta$ will allow us to obtain different PAC-Bayes bounds:

- Let $\Delta(u, v) = v - u$. This is compatible with typical PAC-Bayes bounds on the difference between the true and empirical risks.

- Given $\lambda > 1/2$, let $\Delta$ be defined by $\Delta(u, v) = v - \frac{2\lambda}{2\lambda - 1} u$. This choice is compatible with a Catoni-type bound, as we will see below.

- Let $\Delta$ be defined by $\Delta(u, v) = \mathrm{kl}(u \,\|\, v) = u \ln \frac{u}{v} + (1 - u) \ln \frac{1-u}{1-v}$. This choice is compatible with a Langford-Seeger-type bound. However, note that the function $\Delta$ defined here does not quite obey the hypotheses of lemma 3.1. Indeed, it is only defined for $(u, v) \in [0, 1]^2$, and only has the right monotonicity properties on the set $\{(u, v) \in [0, 1]^2 \mid u \leq v\}$. We can remedy those defects through small adjustments to the proof.

We prove a generalization bound of Catoni-type as an illustration of the machinery just described.

**Theorem 3.2** (Theorem 2 in McAllester [2013]). *Let $\delta \in (0, 1)$ and $\lambda > 1/2$. Fix $i \in [n]$, and let $P_i$ be a probability measure on $\mathcal{H}_i$ (chosen without seeing the training data). Then, with probability at least $1 - \delta$ over the draws of $S$, for all probability measures $Q_i$ on $\mathcal{H}_i$, we have that*

$$R(Q_i) \leq \frac{2\lambda}{2\lambda - 1}\left( R_S(Q_i) + \frac{\lambda}{m}\left( \mathrm{KL}(Q_i \,\|\, P_i) + \ln \frac{1}{\delta} \right) \right).$$

**Theorem 3.3.** *Let $\delta \in (0, 1)$, $\epsilon \geq 0$, and $\lambda > 1/2$. For each $i \in [n]$, let $P_i$ be a probability measure on $\mathcal{H}_i$ (chosen without seeing the training data). Then, with probability at least $1 - \delta$ over the draws of $S$, for all probability measures $Q = Q_1 \otimes \cdots \otimes Q_n$ on $\mathcal{H}$, all $\mathbf{g} \in \mathcal{G}$ that satisfy $\epsilon$-LDP, and all $x' \in \mathcal{X}$, we have that*

$$R(\mathbf{g}, Q) \leq \frac{2\lambda e^{\epsilon}}{2\lambda - 1}\left( e^{\epsilon} R_S(\mathbf{g}, Q) + \frac{\lambda}{m}\left( \mathop{\mathbb{E}}_{i \sim \mathbf{g}(x')} \mathrm{KL}(Q_i \,\|\, P_i) + \ln \frac{n}{\delta} \right) \right).$$

*Proof.* By $n$ applications of Theorem 3.2, we have that, for each $i \in [n]$, with probability at least $1 - \delta/n$, for all $Q_i$,

$$R(Q_i) \leq \frac{2\lambda}{2\lambda - 1}\left( R_S(Q_i) + \frac{\lambda}{m}\left( \mathrm{KL}(Q_i \,\|\, P_i) + \ln \frac{n}{\delta} \right) \right).$$

We can make all these inequalities (for each $i \in [n]$) hold simultaneously with a union bound. Now, applying Lemma 3.1 with $\Delta(u, v) = v - \frac{2\lambda}{2\lambda - 1}u$, we find that, with probability at least $1 - \delta$, for all $Q$, all $\mathbf{g} \in \mathcal{G}$ and all $x' \in \mathcal{X}$, we have that

$$e^{-\epsilon} R(\mathbf{g}, Q) - \frac{2\lambda e^{\epsilon}}{2\lambda - 1} R_S(\mathbf{g}, Q) \leq \mathop{\mathbb{E}}_{i \sim \mathbf{g}(x')}\left( R(Q_i) - \frac{2\lambda}{2\lambda - 1} R_S(Q_i) \right)$$

$$\leq \frac{2\lambda^2}{(2\lambda - 1)m}\left( \mathop{\mathbb{E}}_{i \sim \mathbf{g}(x')} \mathrm{KL}(Q_i \,\|\, P_i) + \ln \frac{n}{\delta} \right). \qquad \square$$

We also give a bound of Langford-Seeger type, since they are generally recognized as among the tightest PAC-Bayes bounds available, and to prove the flexibility of our approach.

**Theorem 3.4.** *Let $\delta \in (0, 1)$, $\epsilon \geq 0$, and $m \geq 8$. For each $i \in [n]$, let $P_i$ be a probability measure on $\mathcal{H}_i$ (chosen without seeing the training data). Then, with probability at least $1 - \delta$ over the draws of $S$, for all probability measures $Q = Q_1 \otimes \cdots \otimes Q_n$ on $\mathcal{H}$, all $\mathbf{g} \in \mathcal{G}$ that satisfy $\epsilon$-LDP, and all $x' \in \mathcal{X}$, we have that, either $R(\mathbf{g}, Q) < e^{2\epsilon} R_S(\mathbf{g}, Q)$, or*

$$\mathrm{kl}(e^{\epsilon} R_S(\mathbf{g}, Q) \,\|\, e^{-\epsilon} R(\mathbf{g}, Q)) \leq \frac{1}{m}\left( \mathop{\mathbb{E}}_{i \sim \mathbf{g}(x')} \mathrm{KL}(Q_i \,\|\, P_i) + \ln \frac{2n\sqrt{m}}{\delta} \right).$$

*Proof.* The proof, which is similar to that of Theorem 3.3, is available in Appendix A. $\qquad \square$

## 3.1 Comparison with other bounds

Very few generalizations bound tailored specifically to mixtures of experts appear in the literature, and those we could find do not apply to mixtures of experts with the one-out-of-$n$ gating mechanism. We can, however, compare our bounds to those obtained by naively applying generic PAC-Bayes generalization bounds to mixtures of experts. In this case, we need to consider classifiers of the form $(Q_{\mathcal{G}}, Q)$, where $Q_{\mathcal{G}}$ is a probability measure on $\mathcal{G}$, and $Q = Q_1 \otimes \cdots \otimes Q_n$ is a probability measure on $\mathcal{H}_1 \times \cdots \times \mathcal{H}_n$ as before. Then, note that

$$\mathrm{KL}(Q_{\mathcal{G}} \otimes Q_1 \otimes \cdots \otimes Q_n \,\|\, P_{\mathcal{G}} \otimes P_1 \otimes \cdots \otimes P_n) = \mathrm{KL}(Q_{\mathcal{G}} \,\|\, P_{\mathcal{G}}) + \sum_{i=1}^{n} \mathrm{KL}(Q_i \,\|\, P_i).$$

This means that a generic PAC-Bayes bound applied to mixtures of experts will depend on the sum of the KL divergences corresponding to the gating functions and each of the experts. Obviously, this sum could be very large. By imposing $\epsilon$-LDP to the gating functions as in our approach, we can eliminate the stochasticity associated to the gating functions, and rid our bounds of the (potentially very large) $\mathrm{KL}(Q_{\mathcal{G}} \,\|\, P_{\mathcal{G}})$ term. Instead, it is $\epsilon$-LDP which controls our gating functions to ensure generalization. Furthermore, our bounds replace the sum of the KL divergences of the experts by a $\mathbf{g}(x')$-weighted average, which means we can have many more experts with almost no penalty from the theoretical point of view. Indeed, our bounds only depend on the number $n$ of experts logarithmically, through the use of the union bound.

## 4 Rademacher bounds for mixtures of experts

Let us start with a slight modification of Lemma 3.1.

**Lemma 4.1.** *We consider mixtures of experts as defined in section 2.1 and provided with the one-out-of-$n$ routing mechanism. Let $\Delta : \mathbb{R}^2 \to \mathbb{R}$ be a convex function that is decreasing in its first argument and increasing in its second argument, and let $\epsilon$ be a nonnegative real number. Then, for any $\mathbf{g} \in \mathcal{G}$ that satisfies the $\epsilon$-LDP property, for any $\mathbf{h} \in \mathcal{H}$, and for any $x' \in \mathcal{X}$:*

$$\Delta\big(e^\epsilon R_S(\mathbf{g}, \mathbf{h}), e^{-\epsilon} R(\mathbf{g}, \mathbf{h})\big) \leq \mathop{\mathbb{E}}_{i \sim \mathbf{g}(x')} \Delta\big(R_S(h_i), R(h_i)\big)$$

*where $R(h_i) = \mathbb{E}_{x \sim \mathcal{D}} \, \ell(h_i(x), y)$ and $R_S(h_i) = \frac{1}{m} \sum_{j=1}^m \ell(h_i(x_j), y_j)$.*

*Proof.* The proof is similar to that of Lemma 3.1 and is provided in Appendix A. $\qquad\square$

Let us now recall the following definition.

**Definition 4.2** (Rademacher complexity). Given a space $\mathcal{H}$ of predictors, a loss function $\ell$, and a data generating distribution $\mathcal{D}$, the Rademacher complexity $\mathcal{R}(\ell \circ \mathcal{H})$ is defined by

$$\mathcal{R}(\ell \circ \mathcal{H}) \;=\; \mathop{\mathbb{E}}_{S \sim \mathcal{D}^m} \mathop{\mathbb{E}}_{\boldsymbol{\sigma}} \sup_{h \in \mathcal{H}} \frac{1}{m} \sum_{j=1}^m \sigma_j \ell(h(x_j), y_j),$$

where $\boldsymbol{\sigma} = (\sigma_1, \dots, \sigma_m)$ is distributed uniformly on $\{-1, 1\}^m$.

Our main theorem will make use of the following well-known risk bound.

**Theorem 4.3** (Basic Rademacher risk bound). *Given a $[0, 1]$-valued loss function $\ell$, with probability at least $1 - \delta$, for all $h \in \mathcal{H}$, we have that*

$$R(h) \; \leq R_S(h) + 2\mathcal{R}(\ell \circ \mathcal{H}) + \sqrt{\frac{2 \ln(2/\delta)}{m}} \, .$$

**Theorem 4.4.** *Let $\delta \in (0, 1)$ and $\epsilon \geq 0$. Given a $[0, 1]$-valued loss function $\ell$, then, with probability at least $1 - \delta$ over the draws of $S$, for all $\mathbf{h} \in \mathcal{H}_1 \times \cdots \times \mathcal{H}_n$, for all $\mathbf{g} \in \mathcal{G}$ that satisfy $\epsilon$-LDP, and all $x' \in \mathcal{X}$, we have that*

$$R(\mathbf{g}, \mathbf{h}) \leq e^\epsilon \bigg( e^\epsilon R_S(\mathbf{g}, \mathbf{h}) + 2 \mathop{\mathbb{E}}_{i \sim \mathbf{g}(x')} \mathcal{R}(\ell \circ \mathcal{H}_i) + \sqrt{\frac{2 \ln(2n/\delta)}{m}} \bigg).$$

*Proof.* By $n$ applications of Theorem 4.3, we have that, for each $i \in [n]$, with probability at least $1 - \delta/n$, for all $h_i \in \mathcal{H}_i$,

$$R(h_i) \leq R_S(h_i) + 2\mathcal{R}(\ell \circ \mathcal{H}_i) + \sqrt{\frac{2 \ln(2n/\delta)}{m}}.$$

We can make all these inequalities (for each $i \in [n]$) hold simultaneously with a union bound. Now, applying Lemma 4.1 with $\Delta(u, v) = v - u$, we find that, with probability at least $1 - \delta$, for all $\mathbf{h} \in \mathcal{H}_1 \times \cdots \times \mathcal{H}_n$, all $\mathbf{g} \in \mathcal{G}$ and all $x' \in \mathcal{X}$, we have that

$$e^{-\epsilon} R(\mathbf{g}, \mathbf{h}) - e^\epsilon R_S(\mathbf{g}, \mathbf{h}) \leq \mathop{\mathbb{E}}_{i \sim \mathbf{g}(x')} \big(R(h_i) - R_S(h_i)\big)$$

$$\leq \mathop{\mathbb{E}}_{i \sim \mathbf{g}(x')} \bigg( 2\mathcal{R}(\ell \circ \mathcal{H}_i) + \sqrt{\frac{2 \ln(2n/\delta)}{m}} \bigg). \qquad\square$$

Note, that the risk bound of Theorem 4.4 depends only on the average Rademacher complexity of the classes of experts instead of the sum of their Rademacher complexities. Note also that, as in the previous section, the complexity of $\mathcal{G}$ does not affect the risk bound. Finally, the risk bound does not hold uniformly for all values of $\epsilon$. However, by the union bound, the theorem holds for any fixed set $\{\epsilon_1, \dots, \epsilon_k\}$ if we replace $\delta$ by $\delta/k$. Consequently, this suggests a learning algorithm that minimizes $R_S(\mathbf{g}, \mathbf{h})$ for $\epsilon \in \{\epsilon_1, \dots, \epsilon_k\}$.

Also note that Lemma 4.1 allows us to obtain risk bounds for mixtures of experts as long as we have bounds on $\Delta\big(R_S(h_i), R(h_i)\big)$ which hold with high probability, whether they are based on Rademacher complexity, margins, VC dimension, or algorithmic stability.

### 4.1 The need to use adaptive experts

Following these theoretical results, we may be tempted to use a gating network satisfying $\epsilon$-LDP to accomplish a learning task all by itself using non-adaptive experts, that is, experts $h_i$ each taking a constant value, no matter the input: $h_i(x) = i$ for all $x \in \mathcal{X}$. In that case, each Rademacher complexity $\mathcal{R}(\ell \circ \mathcal{H}_i)$ is zero and we can show that Theorem 4.4 can become vacuous under reasonable circumstances.

Consider, for example, the binary classification case with the 0-1 loss. In that case, we have two experts $h_{+1}$ and $h_{-1}$ such that $h_{+1}(x) = +1$ and $h_{-1}(x) = -1$ for all $x \in \mathcal{X}$, and a gating network $\mathbf{g} = (g_{+1}, g_{-1})$. Then, the following holds:

$$
\begin{aligned}
R_S(\mathbf{g}, \mathbf{h}) &= \frac{1}{m} \sum_{j=1}^m \mathbb{E}_{i \sim \mathbf{g}(x_j)} \ell_{\text{0-1}}(h_i(x_j), y_j) \\
&= \frac{1}{m} \sum_{j=1}^m \mathbb{E}_{i \sim \mathbf{g}(x_j)} \mathbf{1}(h_i(x_j) \neq y_j) \\
&\geq \frac{1}{m} \sum_{j=1}^m \sum_{i \in \mathcal{I}} e^{-\epsilon} \max_{x' \in \mathcal{X}} g_i(x') \mathbf{1}(h_i(x_j) \neq y_j), \quad \text{with } \mathcal{I} = \{+1, -1\} \\
&= e^{-\epsilon} \frac{1}{m} \sum_{j=1}^m \max_{x' \in \mathcal{X}} g_{-y_j}(x').
\end{aligned}
$$

Under the assumption that the classes are balanced, meaning that the (marginal) probability of a positive label is equal to the (marginal) probability of a negative label, we have the following:

$$
\begin{aligned}
\lim_{m \to \infty} \frac{1}{m} \sum_{j=1}^m \max_{x' \in \mathcal{X}} g_{-y_j}(x') &= \frac{1}{2} \Big( \max_{x' \in \mathcal{X}} g_{-1}(x') + \max_{x' \in \mathcal{X}} g_{+1}(x') \Big) \\
&\geq \frac{1}{2} \max_{x' \in \mathcal{X}} \big( g_{-1}(x') + g_{+1}(x') \big) = \frac{1}{2}.
\end{aligned}
$$

It follows that, in the limit $m \to \infty$, the risk bound of Theorem 4.4 for any $\mathbf{g}$ has a value of at least $e^{\epsilon}/2 \geq 1/2$. Consequently, the risk bound becomes large or even vacuous in this regime, highlighting the importance of having adaptive experts of finite complexity that can drive the empirical risk to zero when they are selected by the gating network.

## 5 Experiments and results

In what follows, we consider mixtures of $n$ linear experts in binary classification tasks. Let $\mathcal{X} = \mathbb{R}^d$ for some positive integer $d$. Let $S$ be a training set of m examples. Each expert, denoted by $h_i$, where $i$ ranges from 1 to $n$, is characterized by a weight vector $\mathbf{w}_i$. Given an input $\mathbf{x} \in \mathcal{X}$, the output of the expert $h_i$ is given by $h_i(\mathbf{x}) = \mathbf{w}_i \cdot \mathbf{x}$. We use the probit loss function $\ell = \Phi$, which can be seen as a smooth surrogate to the 0-1 loss function, when it is used with an argument of the form $\frac{y \mathbf{w}_i \cdot \mathbf{x}}{\|\mathbf{x}\|}$. In this case, $R(\mathbf{g}, Q)$ and $R_S(\mathbf{g}, Q)$ are given by:

$$
R(\mathbf{g}, Q) = \mathbb{E}_{(\mathbf{x}, y) \sim \mathcal{D}} \mathbb{E}_{i \sim \mathbf{g}(\mathbf{x})} \Phi\Big( \frac{y \mathbf{w}_i \cdot \mathbf{x}}{\|\mathbf{x}\|} \Big)
$$

and

$$
R_S(\mathbf{g}, Q) = \frac{1}{m} \sum_{j=1}^m \sum_{i=1}^n g_i(\mathbf{x}_j) \Phi\Big( \frac{y_j \mathbf{w}_i \cdot \mathbf{x}_j}{\|\mathbf{x}_j\|} \Big), \tag{1}
$$

where $\Phi(x) = \frac{1}{\sqrt{2\pi}} \int_x^{+\infty} e^{-t^2/2} \, dt$ provides the probability that a standard normal random variable is greater than a given value $x$.

To illustrate the regularizing effect of the LDP condition, we carried out several experiments, on different datasets, by minimizing the empirical risk as defined in Equation 1. For all experiments, our models consist of mixtures of $n = 100$ linear experts and a gating network. The gating network is a

neural network having 2 hidden layers. It is parameterized by weights $\mathbf{W}_1 \in \mathbb{R}^{64 \times d}$, where $d$ is the dimension of input vectors, $\mathbf{W}_2 \in \mathbb{R}^{64 \times 64}$, and $\mathbf{W}_3 \in \mathbb{R}^{n \times 64}$, and biases $\mathbf{b}_1 \in \mathbb{R}^{64}$, $\mathbf{b}_2 \in \mathbb{R}^{64}$ and $\mathbf{b}_3 \in \mathbb{R}^n$. Given an input $\mathbf{x} \in \mathbb{R}^d$, the output of the gating network $\mathbf{g}(\mathbf{x}) = (g_1(\mathbf{x}), \ldots, g_n(\mathbf{x}))$ is computed as follows: first, we compute $\mathbf{f}_0(\mathbf{x}) = \tanh(\mathbf{W}_2 \operatorname{ReLU}(\mathbf{W}_1 \mathbf{x} + \mathbf{b}_1) + \mathbf{b}_2)$. Then, when we want the $\epsilon$-LDP condition to be satisfied, we ensure that the outputs are between $-\epsilon/4$ and $\epsilon/4$:

$$\mathbf{f}(\mathbf{x}) = \begin{cases} \frac{\epsilon \mathbf{W}_3 \mathbf{f}_0(\mathbf{x})}{4\|\mathbf{f}_0(\mathbf{x})\|\|\mathbf{W}_3\|_F} & \text{if the gating network must satisfy } \epsilon\text{-LDP} \\ \mathbf{W}_3 \mathbf{f}_0(\mathbf{x}) & \text{otherwise.} \end{cases}$$

Note that $\tanh$ is the hyperbolic tangent activation function, ReLU the Rectified Linear Unit function, $\|\mathbf{W}_3\|_F$ the Frobenius norm of the matrix $\mathbf{W}_3$, and $\|\mathbf{f}_0(\mathbf{x})\|$ the euclidean norm of the vector $\mathbf{f}_0(\mathbf{x})$. Indeed, if we let $\mathbf{W}_3^i$ denote the $i$-th row of $\mathbf{W}_3$, then the $i$-th component of $\mathbf{W}_3 \mathbf{f}_0(\mathbf{x})$ is

$$\mathbf{W}_3^i \cdot \mathbf{f}_0(\mathbf{x}) \le \|\mathbf{W}_3^i\|\|\mathbf{f}_0(\mathbf{x})\| \le \|\mathbf{W}_3\|_F \|\mathbf{f}_0(\mathbf{x})\|,$$

by the Cauchy-Schwarz inequality and the definition of the Frobenius norm. The reason we use the Frobenius norm instead of directly using $\|\mathbf{W}_3^i\|$ is to preserve the proportions between the components of $\mathbf{W}_3 \mathbf{f}_0(\mathbf{x})$ when setting up $\epsilon$-LDP.

The final output of the gating network is given by

$$g_i(\mathbf{x}) = \frac{\exp(f_i(\mathbf{x}) + (\mathbf{b}_3)_i)}{\sum_{k=1}^n \exp(f_k(\mathbf{x}) + (\mathbf{b}_3)_k)} \quad \text{for all} \quad i \in [n].$$

In our experiments, we ran the Stochastic Gradient Descent algorithm 10 times with a learning rate fixed to $0.1$. In each experiment, we trained the model for 1000 epochs, except for the MNIST dataset, where the training duration was shortened to 300 epochs due to dataset size. We allocated approximately 75% of the data to the training set and the remaining 25% to the test set. At the outset of each experiment, the weights of our neural networks were reinitialized to ensure a fresh starting point. After each training run, we computed both the training and test loss values to evaluate the model's performance. We first ran the training without imposing any constraints on the gating network, except for the architecture. Then, we ran several experiments with a gating mechanism satisfying $\epsilon$-LDP, with $\epsilon \in \{0.5, 2, 4, 5, 10\}$. A summary of the results is shown in Table 1. One can observe that regularization with $\epsilon$-LDP improves results in practice, and this regularization is even more evident when the models employing a gating network not satisfying LDP overfit heavily, as in the Breast Cancer and Heart experiments. The regularization effect is slightly less pronounced on MNIST, where the overfitting is not as severe as with the previous datasets. We can also observe the importance of choosing the right hyperparameter $\epsilon$. Indeed, if the value is too small, the output of the gating network becomes insufficiently dependent on the input $\mathbf{x}$. In this case, the experts have to do all the work, and the gating network does not allow them to specialize in well-defined subsets of the instance space. This makes our model closer to a weighted sum of linear classifiers and significantly reduces its performance. Conversely, if $\epsilon$ is overly large, our model tends towards a situation where the LDP condition does not hold, making it prone to overfitting.

Note that our experiments are executed on GPUs in order to parallelize computations and take advantage of the sparsity of our model, but they can also be performed without GPUs. The duration of experiments can range from a few minutes for small datasets such as Breast Cancer to around 3 hours for large datasets like MNIST.

## 6   Conclusion

In this work, we introduce a new way to regularize mixtures of experts. We provide both theoretical and algorithmic contributions in this regard. Our approach offers a significant advantage in that it allows us to harness the remarkable performances of neural networks by using them as gating networks, without being constrained by their architecture or their complexity from the theoretical point of view. By imposing LDP, we obtain nonvacuous bounds on the mixture of experts' risk. Our bounds can become significantly tighter than those presented in section 3.1 and those presented in

---

[2]If $N$ denotes the number of runs, $R_k$ denotes the training or test empirical risk during the $k$-th run, and $\bar{R}$ denotes the average, then standard deviation is given by $\sqrt{\frac{1}{N} \sum_{k=1}^N (R_k - \bar{R})^2}$.

Table 1: Experiment results for mixtures of 100 linear models applied to binary classification tasks: Ads, Breast Cancer [Zwitter and Soklic, 1988], Heart [Janosi et al., 1988] and MNIST [Deng, 2012]. The objective is to minimize the empirical risk as defined in Equation 1. We report the mean training loss ($R_S$) and mean test loss ($R_T$), averaged over ten runs, along with their associated standard deviations.[2]

| Dataset | Risk | No LDP | MoE with a gating network satisfying $\epsilon$-LDP | | | | |
| --- | --- | --- | --- | --- | --- | --- | --- |
| | | | $\epsilon = 0.5$ | $\epsilon = 2$ | $\epsilon = 4$ | $\epsilon = 5$ | $\epsilon = 10$ |
| Ads | $R_S$ | 0.02425 | 0.13854 | 0.01829 | 0.05288 | 0.06459 | 0.02811 |
| | $\pm$ | 0.00499 | 0.00261 | 0.00216 | 0.05543 | 0.05821 | 0.03648 |
| | $R_T$ | 0.03822 | 0.13051 | **0.03206** | 0.06693 | 0.07757 | 0.04384 |
| | $\pm$ | 0.00696 | 0.01138 | **0.00564** | 0.05276 | 0.05822 | 0.03501 |
| Breast | $R_S$ | 0.00780 | 0.04520 | 0.01252 | 0.01062 | 0.01089 | 0.01207 |
| Cancer | $\pm$ | 0.00347 | 0.00426 | 0.00182 | 0.00286 | 0.00193 | 0.00181 |
| | $R_T$ | 0.03617 | 0.04930 | 0.03238 | 0.03297 | 0.02942 | **0.02604** |
| | $\pm$ | 0.01505 | 0.01244 | 0.01349 | 0.01379 | 0.00948 | **0.01277** |
| Heart | $R_S$ | 0.00001 | 0.03524 | 0.00015 | 0.00010 | 0.00009 | 0.00013 |
| | $\pm$ | 0.00000 | 0.00487 | 0.00002 | 0.00001 | 0.00001 | 0.00006 |
| | $R_T$ | 0.00029 | 0.03962 | **0.00026** | 0.00026 | 0.00032 | 0.00032 |
| | $\pm$ | 0.00065 | 0.01013 | **0.00014** | 0.00033 | 0.00030 | 0.00032 |
| MNIST | $R_S$ | 0.00525 | 0.00558 | 0.00529 | 0.00504 | 0.00536 | 0.00523 |
| 0 vs 8 | $\pm$ | 0.00029 | 0.00059 | 0.00044 | 0.00031 | 0.00031 | 0.00032 |
| | $R_T$ | 0.00844 | 0.00869 | 0.00815 | 0.00864 | **0.00769** | 0.00802 |
| | $\pm$ | 0.00103 | 0.00109 | 0.00131 | 0.00165 | **0.00144** | 0.00067 |
| MNIST | $R_S$ | 0.00287 | 0.00330 | 0.00289 | 0.00285 | 0.00298 | 0.00286 |
| 1 vs 7 | $\pm$ | 0.00024 | 0.00033 | 0.00028 | 0.00025 | 0.00023 | 0.00013 |
| | $R_T$ | 0.00501 | 0.00485 | 0.00501 | 0.00518 | **0.00450** | 0.00526 |
| | $\pm$ | 0.00042 | 0.00101 | 0.00093 | 0.00098 | **0.00101** | 0.00066 |
| MNIST | $R_S$ | 0.01419 | 0.01509 | 0.01388 | 0.01396 | 0.01440 | 0.01154 |
| 5 vs 6 | $\pm$ | 0.00046 | 0.00057 | 0.00038 | 0.00051 | 0.00056 | 0.00336 |
| | $R_T$ | 0.02195 | 0.02131 | 0.02206 | 0.02236 | 0.02072 | **0.01852** |
| | $\pm$ | 0.00111 | 0.00160 | 0.00185 | 0.00269 | 0.00229 | **0.00518** |

Azran and Meir [2004], especially in cases where the empirical risk is close to zero and $\epsilon < \ln n$. However, as the empirical risk is multiplied by $e^\epsilon$, the bounds can become loose when $\epsilon$ is large and the empirical risk is significant.

Even though the $\epsilon$-LDP condition is easy to set up, a challenge arises in striking a balance between the parameter $\epsilon$ and the KL divergence or the Rademacher complexity of our experts. Our method introduces an extra hyperparameter $\epsilon$ to optimize but does not provide theoretical guidance on configuring it. This forces us to navigate a trade-off between the value of $\epsilon$, which measures the extent to which the output of the gating network can depend on a given $x \in \mathcal{X}$, and the complexity of our experts, which reflects how well our model captures the data distribution. Finding the right balance requires empirical testing and careful consideration and can open up new avenues of study in the future.

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

## A Proofs and auxiliary results

*Proof of theorem 2.2.* Given $x \in \mathcal{X}$, let $Z(x) = \sum_{i=1}^{n} \exp(\beta f_i(x) + c_i)$, for convenience.

For all $x, x' \in \mathcal{X}$ and all $i \in [n]$, we have that

$$\frac{g_i(x)}{g_i(x')} = \exp\big(\beta(f_i(x) - f_i(x'))\big) \frac{1}{Z(x)} \sum_{k=1}^{n} \exp(\beta f_k(x') + c_k)$$

$$= \exp\big(\beta(f_i(x) - f_i(x'))\big) \frac{1}{Z(x)} \sum_{k=1}^{n} \exp(\beta f_k(x) + c_k) \exp\big(\beta(f_k(x') - f_k(x))\big)$$

$$\leq \max_{i \in [n]; \, x_1, x_2 \in \mathcal{X}} \exp\big(2\beta(f_i(x_1) - f_i(x_2))\big) \frac{1}{Z(x)} \sum_{k=1}^{n} \exp(\beta f_k(x) + c_k)$$

$$\leq \exp(4\beta b). \qquad \square$$

**Theorem A.1** (Jensen's inequality, proposition 1.1 in Perlman [1974]). *Let $\Omega$ be a probability space, let $A$ be a convex subset of $\mathbb{R}^k$, let $X : \Omega \to A$ be an integrable vector-valued random variable, and let $\phi : A \to \mathbb{R}$ be a convex function. Then, $\mathbb{E} X \in A$, and $\phi(\mathbb{E} X) \leq \mathbb{E} \phi(X)$ (in particular, the right-hand side of this inequality exists, though it may be infinite).*

**Theorem A.2** (Theorem 5 in Maurer [2004]). *Let $\delta \in (0, 1)$ and $m \geq 8$. Fix $i \in [n]$, and let $P_i$ be a probability measure on $\mathcal{H}_i$ (chosen without seeing the training data). Then, with probability at least $1 - \delta$ over the draws of $S$, for all probability measures $Q_i$ on $\mathcal{H}_i$, we have that*

$$\mathrm{kl}(R_S(Q_i) \,\|\, R(Q_i)) \leq \frac{1}{m}\Big(\mathrm{KL}(Q_i \,\|\, P_i) + \ln \frac{2\sqrt{m}}{\delta}\Big).$$

*Proof of theorem 3.4.* As remarked earlier, the function $(u, v) \to \mathrm{kl}(u \,\|\, v) : [0, 1]^2 \to \mathbb{R}$ does not exactly satisfy the hypotheses of lemma 3.1, but it is convex. Moreover, on $\{\, (u, v) \in [0, 1]^2 \,|\, u \leq v \,\}$, it is decreasing in its first argument and increasing in its second argument. Also note that, assuming that $R(\mathbf{g}, Q) \geq e^{2\epsilon} R_S(\mathbf{g}, Q)$, then we also have the following inequalities:

$$0 \leq \mathop{\mathbb{E}}_{i \sim \mathbf{g}(x')} R_S(Q_i) \leq e^{\epsilon} R_S(\mathbf{g}, Q) \leq e^{-\epsilon} R(\mathbf{g}, Q) \leq \mathop{\mathbb{E}}_{i \sim \mathbf{g}(x')} R(Q_i) \leq 1.$$

It follows that

$$\mathrm{kl}(e^{\epsilon} R_S(\mathbf{g}, Q) \,\|\, e^{-\epsilon} R(\mathbf{g}, Q)) \leq \mathrm{kl}\Big(\mathop{\mathbb{E}}_{i \sim \mathbf{g}(x')} R_S(Q_i) \,\Big\|\, e^{-\epsilon} R(\mathbf{g}, Q)\Big)$$

$$\leq \mathrm{kl}\Big(\mathop{\mathbb{E}}_{i \sim \mathbf{g}(x')} R_S(Q_i) \,\Big\|\, \mathop{\mathbb{E}}_{i \sim \mathbf{g}(x')} R(Q_i)\Big),$$

and therefore

$$\mathrm{kl}(e^{\epsilon} R_S(\mathbf{g}, Q) \,\|\, e^{-\epsilon} R(\mathbf{g}, Q)) \leq \mathop{\mathbb{E}}_{i \sim \mathbf{g}(x')} \mathrm{kl}\big(R_S(Q_i) \,\|\, R(Q_i)\big)$$

by Jensen's inequality. Now, by theorem A.2, for a fixed $i$, with probability at least $1 - \delta/n$, we have that

$$\mathrm{kl}\big(R_S(Q_i) \,\|\, R(Q_i)\big) \leq \frac{1}{m}\Big(\mathrm{KL}(Q_i \,\|\, P_i) + \ln \frac{2n\sqrt{m}}{\delta}\Big).$$

We can make the above inequality hold for all $i \in [n]$ simultaneously with the union bound. Then, with probability at least $1 - \delta$, for all $(\mathbf{g}, Q)$, given that $R(\mathbf{g}, Q) \geq e^{2\epsilon} R_S(\mathbf{g}, Q)$, we have that

$$\mathrm{kl}(e^{\epsilon} R_S(\mathbf{g}, Q) \,\|\, e^{-\epsilon} R(\mathbf{g}, Q)) \leq \frac{1}{m}\Big(\mathop{\mathbb{E}}_{i \sim \mathbf{g}(x')} \mathrm{KL}(Q_i \,\|\, P_i) + \ln \frac{2n\sqrt{m}}{\delta}\Big). \qquad \square$$

*Proof of Lemma 4.1.* Since the gating function satisfies $\epsilon$-LDP, we have that $e^{-\epsilon} g_i(x') \leq g_i(x) \leq e^{\epsilon} g_i(x')$ for all $x, x' \in \mathcal{X}$ and all $i \in [n]$. It follows that $e^{\epsilon} R_S(\mathbf{g}, \mathbf{h}) \geq \mathbb{E}_{i \sim \mathbf{g}(x')} R_S(h_i)$ and $e^{-\epsilon} R(\mathbf{g}, \mathbf{h}) \leq \mathbb{E}_{i \sim \mathbf{g}(x')} R(h_i)$. Given that $\Delta$ is decreasing in its first argument and increasing in its second argument, we find that

$$\Delta\big(e^{\epsilon} R_S(\mathbf{g}, \mathbf{h}), e^{-\epsilon} R(\mathbf{g}, \mathbf{h})\big) \leq \Delta\Big(\mathop{\mathbb{E}}_{i \sim \mathbf{g}(x')} R_S(h_i), \mathop{\mathbb{E}}_{i \sim \mathbf{g}(x')} R(h_i)\Big)$$

Since $\Delta$ is a convex function, we can apply Jensen's inequality to the expression on the right-hand side, yielding the desired result. $\qquad \square$

