# OpenReview forum: "Local Differential Privacy for Mixtures of Experts"
_NeurIPS.cc/2024/Conference — Submitted to NeurIPS 2024_

### Official Review · Reviewer_Cnn1 · 2024-07-11

**Soundness:** 2
**Presentation:** 2
**Contribution:** 3
**Rating:** 3
**Confidence:** 3

**Summary:**

This paper introduces a novel mixture of experts model that applies local differential privacy to the gating mechanism. Their methods leverages the one-out-of-n gating mechanism and provides specific generalization bounds.

**Strengths:**

1. The overall insight of the paper is clear and strong.
2. Improve the tightness of bounds on the risk for mixtures of experts models.
3. Reduce complexity by relying on fewer parameters.

**Weaknesses:**

1. The section 3 talks about PAC-Bayesian bounds for mixtures of experts, but it lacks insight into why PAC-Bayesian bounds are applied instead of other bounds. More explanation is needed here, similar to the explanation needed in section 4 regarding Rademacher bounds.

2. In the experiment section, it is unclear why the chosen dataset is used for the experiments and why only 5 epsilon values were selected.

3. Even though there are very few existing guarantees, the experiment should include other methods as baselines and compare the results.
4. For the experiment section, only consider mixtures of n linear experts in binary classification tasks seems easy. Need to add other classification tasks.
5. There is no description about the datasets used in experiments.

**Questions:**

1. For section 3, why there is no reference for PAC-Bayes theory when you first mention it?
2. Why there is no reference about Kullback-Leibler divergence?

**Limitations:**

1. Lack many reference as I motioned in questions part.
2. The paper lacks a smooth flow, making it difficult to follow. Specifically, there is no clear insight or reasoning provided to explain why the existing mechanisms or bounds were chosen, as highlighted in weakness 1.

---

> ### Author Rebuttal · Authors · 2024-08-06
>
> We appreciate the reviewer’s feedback. Here, we address the concerns raised and provide clarifications to improve the overall evaluation of our work.
>
> 1. Regarding the choice of PAC-Bayesian bounds discussed in Section 3, we would like to clarify that PAC-Bayesian bounds have been extensively studied in recent literature and are known for providing tight theoretical guarantees on the risk of various models. This makes them particularly suitable for our analysis. The decision to use PAC-Bayesian bounds was motivated by their ability to offer robust generalization guarantees in probabilistic models, which aligns well with our focus on examining MoE models under LDP. We will revise Section 3 to provide a more detailed explanation of why PAC-Bayesian bounds were selected over other methods, including a discussion of recent advancements and their relevance to our work.
> 2. Similarly, in Section 4, we will expand on the use of Rademacher bounds. We will clarify that Rademacher bounds are employed to compare our results with existing studies, such as the one by Azran et al. cited in the paper. This comparison will help elucidate the role of Rademacher bounds in our analysis and justify their application. Additionally, we will note that our method can be extended to various other bounds, including mutual information-based bounds and others. This will be included in the revised version to provide a broader context for our theoretical approach.
> 3. Concerning the experiments conducted, they are primarily designed to demonstrate that the conditions imposed by our method are reasonable and that the chosen epsilon values are consistent with the theoretical framework. The aim is to validate the practicality of applying LDP within MoE models and to ensure that it does not adversely affect model performance for reasonable privacy parameters. Our main contribution is theoretical, focusing on presenting bounds that are significantly tighter than existing ones. We acknowledge that the current title and abstract may be misleading, and we propose to revise them to more accurately reflect the theoretical nature of our contributions and clarify any potential confusion.
> 4. Also, we will include a reference to the foundational PAC-Bayesian theory when it is first introduced in Section 3, to ensure that readers have a clear understanding of the theoretical background as well as a reference to the Kullback-Leibler divergence in the relevant section of the paper.

---

### Official Review · Reviewer_4Nc6 · 2024-07-13

**Soundness:** 2
**Presentation:** 1
**Contribution:** 2
**Rating:** 4
**Confidence:** 4

**Summary:**

This paper introduces a novel approach to regularize mixtures of experts by imposing local differential privacy (LDP) on the gating mechanism. The authors provide theoretical justifications and derive PAC-Bayesian and Rademacher bounds tailored to this approach. Experiments conducted on various datasets demonstrate that using LDP as a regularizer improves the generalization ability of the models, especially in cases prone to overfitting. The method offers a balance between leveraging neural networks for gating and maintaining robust theoretical guarantees, making it a valuable contribution to the field of machine learning.

**Strengths:**

This paper demonstrates originality by integrating local differential privacy (LDP) into the mixture of experts model, addressing privacy concerns while improving model generalization. The theoretical contributions, including PAC-Bayesian and Rademacher bounds, are rigorously derived and tailored to the new approach. The clarity of exposition makes complex concepts accessible, and the experiments validate the practical benefits of the method. The significance lies in enhancing the robustness and scalability of mixture of experts models, making them more applicable to real-world scenarios prone to overfitting.

**Weaknesses:**

The primary concern regarding this paper lies in its significance. There have been previous works that incorporated differential privacy (DP) into the construction of mixture of experts models with privacy considerations. This paper, however, utilizes local differential privacy (LDP) to analyze the theoretical aspects of mixture of experts models. The introduction of LDP significantly alters the generalization behavior of these models because both LDP and DP are methods that inherently enhance algorithm robustness, thereby affecting generalization. If the main goal of the paper is to enhance privacy, it is imperative to compare this approach with existing DP-based methods and highlight what specific aspects LDP protects that traditional DP cannot. Without this comparison, the added value of using LDP over existing DP methods remains unclear. On the other hand, if the focus is on analyzing the generalization of mixture of experts models, the paper must justify the rationale behind incorporating LDP for this analysis, as LDP is not inherently required for mixture of experts models. The paper needs to elaborate on why LDP is a suitable and necessary tool for this analysis and how it fundamentally impacts the generalization properties of the models in a meaningful way. Additionally, while the theoretical contributions are substantial, the practical implications need to be demonstrated more robustly through experiments. Comparing the results directly with models using traditional DP methods would strengthen the paper by showing the practical improvements and specific scenarios where LDP outperforms DP. Furthermore, using a broader range of datasets could better illustrate the claimed benefits in robustness and scalability. By addressing these concerns, the paper can more convincingly argue the necessity and advantages of using LDP in mixture of experts models, thereby enhancing its significance in the field.

**Questions:**

Please refer to weakness.

**Limitations:**

No.

---

> ### Author Rebuttal · Authors · 2024-08-06
>
> We appreciate the reviewer’s insightful feedback regarding the significance of our paper.
> 1. We acknowledge that there have been previous works incorporating differential privacy (DP) for regularization. Our paper, however, focuses on utilizing local differential privacy (LDP) on a part of the model (the gating network) to analyze the theoretical aspects of MoE models, with a particular emphasis on the generalization properties under LDP conditions. The strength of our bounds lies in the fact that LDP makes them tighter and ensures they depend logarithmically on the number of experts $n$, whereas existing bounds typically depend linearly on  $n$. We leverage the one-out-of-$n$ mechanism to achieve these tighter bounds.
>
> 2. While traditional DP methods also add noise to regularize the model, they do so in a "global" manner. In contrast, our approach imposes LDP specifically on the gating network. We believe that global noise would result in looser bounds that, once again, depend linearly on $n$. Therefore, we believe our method of applying LDP locally is crucial for maintaining the tightness of the bounds.
>
> 3. Also, we are interested in understanding why the presentation was rated as 1 (poor). This feedback would help us improve the quality of our future presentations. We would greatly appreciate it if you could provide specific details or suggestions so that we can better prepare our work moving forward.

---

> > ### Comment · Reviewer_4Nc6 · 2024-08-14
> >
> > I thank the authors for their efforts in clarifying and revising their manuscript.
> >
> > That said, I will keep my original score. I do not feel that the rebuttal is able to resolve my concerns in a meaningful way.
> >
> > I'm happy to discuss my opinion with other reviewers and area chairs.

---

### Official Review · Reviewer_FV3i · 2024-07-13

**Soundness:** 2
**Presentation:** 4
**Contribution:** 2
**Rating:** 4
**Confidence:** 2

**Summary:**

This paper provides generalization bounds for a particular type of mixture of experts (MoE) networks. They focus on MoE architectures where an input $x$ first goes through a gating function $g$, and then gets routed to a single (one out of n) expert $i \in [n]$ according to the $g(x) \in [0,1]^n$ distribution. The final output of the MoE network is the output of the expert $h_i(x)$.

The authors observe that when the gating function $g$ has certain regularization properties, which correspond to local differential privacy (LDP), then the resulting network has better generalization bounds than what appears in the existing MoE litterature. The authors provide such bounds.

Finally, the authors evaluate LDP-regularized routing on binary classification tasks with mixtures of linear models. The results show that LDP regularization outperforms an un-regularized baseline.

**Strengths:**

Mixture of experts models are still understudied, especially from a theoretical standpoint, so I appreciate the new analysis provided by this paper. The connection with differential privacy is creative and seems fruitful, although I have some reservations about it (see below).

The bounds do improve on existing generic bounds for this type of MoE models. The paper is well-written and easy to follow, and the experimental code is available.

**Weaknesses:**

First, it is worth emphasizing that the MoE networks in this paper do not satisfy local differential privacy themselves. The gating network is not even *trained* with differential privacy. This paper only uses local differential privacy as a regularization condition on the *gating network only* at *inference* time, which does not provide meaningful privacy guarantees. That is not immediately clear from the title of the paper. To be fair, this work does not try to achieve any privacy goals, and is entirely focused on generalization bounds. But if privacy is not needed, then it is not clear why DP is the right tool for the job. The paper directly uses LDP (it could have been called something like "exponentially regularized routing") but does not motivate this choice. Are there other forms of regularization that could achieve similar or better bounds? While there are some known connections between robustness, differential privacy, and generalization, they are not mentioned here. In the context of MoEs, some large models already regularize their gating functions (e.g., the Switch Transformer adds some "jitter" noise to the routing logits).

Next, the paper motivates the study of MoE models by mentioning recent progress with LLMs such as the Switch Transformer, which uses multiple layers of experts (deep MoE) and combines the output of different experts. All the modern LLM MoE models I am aware of are such deep MoEs. Meanwhile, the paper focuses on simple shallow MoE models with a single gating network followed a single layer of experts, which limits the potential impact of the paper in my opinion. While theoretical bounds may be of interest even on shallow MoE models, I would appreciate at least some discussion about whether the authors' approach can generalize to deeper models.

Finally, the experiments have some limitations, which mostly stem from the two previous concerns.
* The authors only evaluate a single, rather simplistic (3-layer MLP gating network followed by linear experts), MoE architecture. More concerningly, they use a fixed number of experts (n = 100), thereby missing an opportunity to evaluate their claim that "we can have many more experts with almost no penalty from the theoretical point of view".
* The only baseline is "No LDP", which I think is a quite weak baseline. It is not entirely surprising that adding some regularization, in the form of LDP routing, improves generalization compared to a completely un-regularized baseline. How about other forms of regularization, such as dropout, clipping, or jitter noise (which already exists in the context of MoEs)?
* Another baseline would be a non-MoE model, with a comparable number of parameters, e.g., even a simple, dense, multi-layer perceptron. Showing that shallow MoEs outperform dense models would alleviate concerns about the practical relevance of this work.

Minor comments:
* Table 1 might be more readable as a graph.
* It is quite surprising to see MNIST being qualified as a "large" dataset, for which a 4-layer network takes 3 hours to train on a GPU, in 2024.
* Also, it is unclear why MNIST has to be broken down into 3 binary classification tasks.

**Questions:**

Have you considered other forms of regularization (including the noisy routing techniques that are already used in practice by MoE)? Is there any justification why local differential privacy would be the type of regularization that gives the most desirable generalization bounds?

**Limitations:**

The authors adequately addressed the limitation they identified (difficulty of tuning epsilon), even though this is not the main limitation of this work in my opinion.

---

> ### Author Rebuttal · Authors · 2024-08-06
>
> We would like to thank the reviewer for their thoughtful and detailed feedback. Your insights have highlighted several important areas for improvement and clarification in our work. We appreciate the opportunity to address the concerns raised.
>
>    1. The critique regarding the use of local differential privacy (LDP) as a regularization technique rather than as a strict privacy guarantee is well taken. Our primary contribution is the application of LDP specifically to the gating network within the mixture of experts (MoE) framework. This targeted use of LDP allows us to derive tight theoretical bounds on the model’s risk, which supports the impressive empirical performance of MoE models.  We acknowledge that the title and abstract could better reflect this emphasis on generalization. We will revise these sections in future versions to more clearly convey our focus on theoretical bounds and generalization, rather than privacy. However, it is important to mention that the LDP condition can be achieved by ensuring that each expert is also $\epsilon$-LDP. In this case, the overall model would then provide  $2 \epsilon$-LDP due to the composition of privacy guarantees.
>
> 2. Regarding the baselines, while it would indeed be beneficial to compare LDP with other regularization methods such as dropout, clipping, or jitter noise, our current bounds are specifically tied to the LDP condition. Incorporating other forms of noise might not yield bounds with the same tightness and may not align with the theoretical analysis we have established. This is an interesting direction for future research, and we plan to explore how these alternative methods impact generalization bounds in subsequent studies.
>
> 3. The focus on shallow MoE models was intentional, aimed at providing clear insights into the theoretical bounds. However, we recognize the value in extending our analysis to deeper MoE architectures, such as those used in modern large language models (LLMs) like the Switch Transformer. We are planning follow-up studies to investigate whether our approach generalizes to more complex and deeper MoE models and will include discussions on this in future revisions.

---

> > ### Comment · Reviewer_FV3i · 2024-08-11
> >
> > Thank you for the answer. I appreciate the clarification regarding the title and the use of differential privacy.
> >
> > While composing LDP experts with a LDP gating function does indeed give LDP guarantees for the whole network, these guarantees only hold for a single inference. Since inference-time DP is rarely used and less relevant than DP training of neural networks, I think that this is an interesting side remark but not a particularly strong contribution from the paper, so I agree with the new framing/title/abstract you propose.
> >
> > I still believe that other baselines or more context about why LDP regularization is a natural choice would strengthen the paper, but I am no longer strongly opposed to accepting this paper.

---

> > > ### Author Response · Authors · 2024-08-13
> > >
> > > Thank you for your insightful feedback. We acknowledge your remark regarding the baselines, and we will compare our regularization method with other techniques such as dropout and adding noise in future revisions.
> > >
> > > Regarding the use of the LDP condition, we recognize the importance of justifying our choice and would like to provide some clarification: Our motivation was to find a middle ground between input-independent mechanisms, such as weighted majority vote classifiers, which offer tight theoretical guarantees on risk, and input-dependent aggregation mechanisms, such as the best-known versions of mixtures of experts, which excel in practical performance. Our analysis of existing bounds on mixtures of experts revealed that they can be loose in certain cases—particularly when the weights provided by the gating network do not depend, or depend only *minimally*, on the input. We aimed to bridge these two approaches by introducing a unified framework that quantifies this dependence and incorporates it into the theoretical bounds.

---

### Official Review · Reviewer_StUb · 2024-07-14

**Soundness:** 3
**Presentation:** 3
**Contribution:** 2
**Rating:** 4
**Confidence:** 3

**Summary:**

The authors consider the mixtures of experts models, in particular the one-out-of-n gating mechanism for ease of theoretical analysis, and show that applying a soft-max, which is also the exponential mechanism, on the gating mechanism gives LDP and can improve generalization. The privacy techniques are largely the same as previous work, PATE, but specifically applied to mixtures of experts. The authors then provide theoretical analysis showing generalization bounds for this approach.


Unfortunately, I’m not familiar enough with the mixture of experts literature to evaluate the novelty of applying the soft-max and the corresponding theoretical guarantees. I am rather surprised though that the soft-max has never been applied and am still somewhat confused upon what the previous techniques were in the one-out-of-n mixtures of experts.

**Strengths:**

The authors show that choosing an expert through the soft-max provides better generalization. The privacy guarantee is then essentially proportional to the regularization factor (\beta) for the soft-max application. Further they give theoretical generalization bounds for this approach.

**Weaknesses:**

Unless I am mistaken (please correct me if I’m wrong) the authors are not providing privacy guarantees for the model itself, but only one inference call to the model. In particular, if feature vector x is input to the model, then the expert is chosen randomly according to the soft-max / exponential mechanism, which is \epsilon-LDP. If inference was then run again, suppose even on the same feature vector, then the random draw from experts would occur again. This is known as composition in the privacy literature, and the privacy guarantees	would now be 2*\epsilon.

Providing privacy guarantees on only one inference call to a model is both not very useful nor interesting due to the composition properties.

**Questions:**

See weaknesses section.

**Limitations:**

See weaknesses section.

---

> ### Author Rebuttal · Authors · 2024-08-06
>
> We thank the reviewer for providing valuable feedback. We recognize the need to clarify our main contributions. The softmax function and LDP are well-known techniques that have been used in previous works. However, we believe that our contributions are distinct and significant since they use these techniques as an original way of getting theoretical guarantees on the risk of our models. Here are some explanations regarding the review:
>
> 1. The reviewer is correct in noting that while our gating networks satisfy the
> $\epsilon$-LDP condition, our models do not inherently guarantee this privacy level since we do not impose constraints on the experts. However, LDP of the whole model can be achieved by ensuring that each expert is also
> $\epsilon$-LDP. Under these circumstances, the overall model would then provide
> $2 \epsilon$-LDP due to the composition property. Our innovation is in applying LDP specifically to the gating network within the mixture of experts. This strategic application allows us to derive tight theoretical bounds on the model's risk, substantiating the impressive empirical performance of mixtures-of-experts models.
>
> 2. We realize that the title and abstract of our paper may have led to some confusion regarding our contributions. We will revise these sections to better reflect the novelty and significance of our theoretical analysis and the specific application of LDP in gating networks.
>
>
> We hope these clarifications help convey the originality and scientific value of our work.

---

> > ### Comment · Reviewer_StUb · 2024-08-12
> >
> > Thanks to the authors for their comments and clarifications! Regarding point 1), I can appreciate that this can be part of ensuring LDP, but ensuring the each expert is LDP is the much harder difficulty here, not the exponential mechanism for choosing the expert. Further, the idea of using exponential mechanism or noisy max on \epsilon-DP (or LDP) experts/models to choose one for inference is not a new concept at all. Also the \epsilon parameter here is essentially just acting as a regularizer for the softmax function which is also not a novel concept.
> >
> > The authors have some nice novel theoretical contributions! But I think the necessary re-write is too substantial at this time and in my opinion the connection to LDP is more of a remark given the well understood connection between exponential mechanism and softmax with a regularizer. I will keep my score.

---

### Official Review · Reviewer_YLRL · 2024-07-20

**Soundness:** 3
**Presentation:** 2
**Contribution:** 2
**Rating:** 4
**Confidence:** 4

**Summary:**

The authors  take the first step (I though so at first) to MoE under LDP theoretically.  I read again and found that the author seems to have raised the utility lower bound of existing studies. Few experiments could be found. Perhaps, I am not an expert in MoE, but it really leave a  hard time.

**Strengths:**

Important poblem.

**Weaknesses:**

1. Perhaps I am not an expert in MoE, and I cannot tell from the author's introduction that there are any challenges.
2. The experimental results and application scenarios are not clear.

**Questions:**

1. According to my rough understanding, the proposed approach seems to have some similarity with ensemble learning So what is the difference between the proposed approach and existing  ensemble learning under LDP?
2. MoE seems to be widely used in deep learning, so what are the differences between the proposed method and DP-SGD or LDP-SGD? Is there LDP-SGD-MoE?

**Limitations:**

see weakness.

---

> ### Author Rebuttal · Authors · 2024-08-06
>
> We thank the reviewer for reading our submission and providing valuable feedback. We believe that we should have emphasized more that our main contribution lies in the theoretical risk bounds presented in the article, which can be much tighter than existing bounds. In light of the review, we realize that the title and abstract of our paper are confusing and that certain points regarding our contribution have to be clarified.
>
> 1. To the best of our knowledge, there are no works that impose local differential privacy on gating networks in mixtures of experts or comparable models. Our method bridges the gap between two existing ensemble methods: one that aggregates the outputs of experts using a mechanism that does not depend on the input, and another that aggregates the outputs of predictors using an input-dependent mechanism without any restrictions on this dependency. Typical mixtures of experts are an example of the latter case. Our method is a generalization of these two methods, since the first case can be obtained by setting $\epsilon$ to 0 and the second can be obtained for an $\epsilon$ that tends to infinity.
> 2. Existing methods such as DP-SGD and LDP-SGD share similarities with our methods in the sense that they add noise to satisfy privacy conditions or to regularize models. However, in our case, we impose this condition on only one part of our model, which is the gating network. We use this well-known technique (LDP) as a tool to obtain theoretical bounds on the risk of our models in order to support the impressive empirical results of Mixture of Experts.

---

> > ### Comment · Reviewer_YLRL · 2024-08-14
> >
> > Thks for your response. I would keep my score.

---

### Author Rebuttal · Authors · 2024-08-06

We would like to thank the reviewers for their thorough analysis and valuable feedback on our paper. The comments have allowed us to take a step back and reflect on our work, especially regarding the presentation of our contributions. The reviews have made us realize that our title and abstract are misleading and should be changed to make it clear that our article is about theoretical guarantees on the risk of mixtures of experts.

Indeed, very little theoretical work has been done on mixtures of experts, and we wanted to provide a mathematical analysis to back up the impressive empirical results achieved by mixtures of experts. Note that the existing generalization bounds depend linearly on the number of experts, whereas ours only depend on the logarithm of the number of experts, which makes them much tighter than the existing bounds, under reasonable conditions.

Our guarantees on the risk of mixtures of experts are obtained by imposing local differential privacy (LDP) on the gating network of our models. The motivation for this is that imposing LDP amounts to making the outputs of the gating network less dependent on the particular example being classified, which can be seen as a way of controlling its complexity. This has the benefit of entirely eliminating the complexity/KL divergence term associated to the gating network from our risk bounds; instead, generalization is controlled by the parameter $\epsilon$. This novel approach bridges the gap between two existing ensembling methods: one that uses an input-independent aggregation mechanism, as in weighted majority vote classifiers, and another that employs an input-dependent mechanism without any constraints regarding the dependence between the input and the output of the gating mechanism, as in the best-known mixtures of experts models. Our method generalizes these approaches, with
$\epsilon = 0$ replicating the first method and
$\epsilon \to \infty$ replicating the second.

Note that LDP is just a formal condition we have found helpful in our theoretical analysis, and we are *not* claiming that our models satisfy any privacy guarantees, nor was this the point of our work. We were only looking to understand the generalization of mixtures of experts by providing bounds on their risk. However, LDP can be satisfied by the whole model by ensuring that our experts satisfy $\epsilon$-LDP. In this case, the mixture of experts would satisfy $2\epsilon$-LDP. To clarify this point, we propose including this explanation in Section 2.2.

Also, please note that our experiments have been conducted solely to support our theory and to show that the conditions imposed on our gating networks are reasonable. As we showed, in addition to providing theoretical guarantees, the imposition of LDP on the gating network does not deteriorate the quality of the learning, but rather acts as a form of regularization. The aim of this paper was not to beat state of the art mixtures-of-experts models. Instead, our aim was to take steps toward a theoretical analysis of their performance.

For the reasons mentioned above, we suggest changing the name of our article to *Tighter Risk Bounds on Mixtures of Experts* and editing our abstract as follows:

*In this work, we provide theoretical guarantees on the risk of mixtures of experts by imposing local differential privacy on their gating mechanism. These bounds are specifically tailored for mixtures of experts provided with the one-out-of-$n$ gating mechanism rather than the more conventional $n$-out-of-$n$ mechanism and depend on the number of experts only logarithmically. This makes them much tighter than the existing bounds, under reasonable conditions. Experimental results support our theory, demonstrating that our approach enhances the generalization capability of mixtures of experts and validates the feasibility of the imposed conditions.*

---

### Decision · Program_Chairs · 2024-09-25

**Decision:**

Reject

**Comment:**

This paper proposes a new regularization for mixture of experts (MoE) model based on the ideas from differential privacy. In particular, the gating function (which assigns probabilities to the experts) is forced to satisfy $\epsilon$-local differential privacy ($\epsilon$-LDP), meaning that changing the input does not change each probability by more than a factor of $e^{\epsilon}$. The paper shows that this regularization allows one to prove generalization bounds for MoEs, and that, empirically, it also performs well (compared to no such regularization) on shallow NNs.

Although this paper tackles an interesting problem, the current paper's presentation leaves a majority of the reviewers confused. Namely, the proposed method does *not* yield any meaningful privacy guarantee, but rather is purely to give generalization bounds. This should be highlighted more clearly both in the title and the main paper text. Furthermore, no discussion is given as to why this regularization is used as opposed to others; similarly, no other baselines are used in the experiments. With a major revision, this paper could be a very good contribution to the field. However, as is, the paper is not yet ready to be published at NeurIPS.